# A retrospective analysis thyroid function and ultrasonography in a group of subjects with lepramatous leprosy in Turkey

**Zeynep Ozkan**[1]*, **Zekiye Kanat**[2], **Ozkan Alatas**[3], **Zuhal Karaca Karagoz**[4]

**1** Usak University, Medical Faculty, Department of General Surgery, Usak, Turkey, **2** Malatya Turgut Özal University, Faculty of Medicine, Department of Dermatology, Malatya, Turkey, **3** Dokuz Eylül University, Faculty of Medicine Department of Radiology, Izmir, Turkey, **4** Health Sciences University, Elazig Fethi Sekin City Hospital, Department of Endocrinology, Elazig, Turkey

* drzeynepozkan@yahoo.com

**Data Availability Statement:** All data are in the manuscript and/or supporting information files.

**Funding:** The author(s) received no specific funding for this work.

## Abstract

### Background

There are several studies on thyroid functions and thyroid gland features in patients with leprosy in the literature. The relationship between them have not been clarified yet. These studies are time-expired and don't contain ultrasonography examination. The purpose of the study is to investigate thyroid functions and gland characteristics in leprosy patients by ultrasonography (US) and current laboratory techniques.

### Patients and Methods

This retrospective study was conducted by collecting the data of patients with lepramatous leprosy. Serum thyroid-stimulating hormone, free triiodothyronine, free thyroxine, anti-thyroid peroxidase, antithyroglobulin, and thyroglobulin values and thyroid ultrasonography reports were collected from previous records.

### Results

The mean age is 75.12±9.89 years of total 17 subjects and 10 patients (58.8%) were male. Thyroid US was performed on 14 of the patients, nodules were detected in a total of 7 (50%) patients. The mean FT3, FT4, TSH, Anti-Tpo, Anti-TG, and TG values of the patients were found to be within normal limits.

### Conclusion

In the present study, no changes were detected in the thyroid functions and structures of the patients with Lepromatous Leprosy. We consider that prospective randomized studies that will include larger sample sizes are needed to determine whether there is a relationship between leprosy and thyroid disease.

**Competing interests:** The authors have declared that no competing interests exist.

## Author summary

Leprosy is also known as Hansen's disease, caused by the bacillus Mycobacterium leprae. This ancient disease continues to be a serious healt problem worldwide. Currently it is more common in subtropical regions. Although leprosy cases are decreasing, they are still seen in Turkey. Mycobacterium Leprae lead to an chronic granulamatous infection mainly on skin and peripheral nerves however, it can also affect other tissues such as eyes, mucous membranes, bones and testicles, and may occur different clinical manifestations. It has been shown that it also causes changes in endocrine organ functions. There are some studies on thyroid functions, but the effects of leprosy on thyroid functions and structure are not clear.

In this study, we examined the structural features and functional status of the thyroid gland, in a small number of patients because of Leprosy is non endemic common in our country.

No changes were detected in the thyroid functions and structures of the patients with Lepromatous Leprosy in a thyroid-endemic region in this study. We consider that it is necessary prospective randomized studies included larger sample sizes on this topic

## Introduction

Leprosy is a chronic granulomatous infectious disease caused by a bacillus called Mycobacterium Leprae, and characterized by severe dermatological and neurological disabilities. There are no known reservoirs other than humans. Although the causative agent of leprosy, which is one of the oldest known diseases of human history, was first described in 1873, there are some unknown points in information about its spread [1,2].

Leprosy has occasionally been observed in almost every part of the world. Today, it is more frequently seen in subtropical regions, most commonly in India, Brazil, Indonesia, Bangladesh, and Ethiopia, and rarely in Europe and our country, Turkey. While its annual incidence is approximately 250.000–500.000 cases in the world, there were about only 1.200 patients in Turkey in 2012 according to the records. The annual number of new cases has been reported to be between 1–5 in recent years. Residing in an endemic region, family history and poverty are among the risk factors of leprosy, which can be seen at any age. When the bacillus, which is mostly eliminated by the body, causes the infection, tuberculoid leprosy, borderline leprosy, and lepramatous leprosy emerge in various clinicopathological clinics depending on the immune response [2]. M. Leprae, which infects the mucosa, peripheral nerves and reticuloendothelial tissues, causes permanent neurological deficits in some patients even if treated, and traumatic ulcers are observed in the extremities due to the loss of senses [3]. In addition, leprosy, which genetically has a pleiotropic characteristic, has been reported to affect many different organs, and demonstrated to cause changes in endocrine organ functions [4].

While there are some studies on thyroid functions, the results related to the effects of leprosy on thyroid functions and structure have not been clarified yet [4]. There are various studies to examine thyroid functions in leprosy patients, and conflicting results have been found in previous studies regarding thyroid functions and the disease.Current laboratory tests and imaging methods are more sensitive for evaluating the thyroid disorders than in the past. We aimed in this study to investigate thyroid functions and gland morphology and tissue characteristics of thyroid gland in leprosy patients using Ultrasonography (US) and laboratory techniques (Electrochemiluminescence Immunoassay (ECLIA), for which Radioimmunoassay

(RIA) Test was used formerly. It was considered that these modern methods can help to understand whether there is a relationship between leprosy and thyroid disorders.

In the literature, thyroid ultrasonography the first imaging technique recommended in guielines for thyroid diseases, was never used in leprosy patients previously. Inthis study, the purpose was to present thyroid functions and ultrasonographic characteristics of leprosy patients. A brief on what the research problem is the justification of this study even if it is a retrospective study as this will give a reason for future study.

## Materials and Methods

### Ethics statement

The ethics committee approval for the study, dated 29.04.2020 and numbered 2020/07-25, was obtained from the Non-Invasive Research Ethics Committee of Firat University. All procedures performed in the study involving human participants were in accordance with the ethical standards of the institutional and/or national research committee and with the 1964 Helsinki Declaration and its later amendments or comparable ethical standards.

This study was retrospectively conducted on inpatients who were previously diagnosed with lepramatous leprosy in the Leprosy Clinic of Elazig Training and Research Hospital. These patients were diagnosed by these criteria; detection of acid-fast bacilli in slit skin smears, and the histopathological examination of skin biopsies and typical dermal appearances caused by thickened nerves.

The patients who were included in the study were previously diagnosed in the Leprosy Ward of our hospital and were followed for a long time. This ward is a continuation of the leprosy hospital, which was established during the leprosy fight that started approximately 80 years ago in our country, and the patients in this ward are those with chronic follow-up. The patients are not currently receiving active treatment and are only receiving social care services because of the sequelae of the disease. These patients had previously been treated for leprosy and had their thyroid tests and ultrasonograpic examination done during the follow-ups.

The fight against leprosy has been effectively performed in specialized leprosy hospitals in our country since 1983 and the number of patients has decreased a lot. After reaching the World Health Organization leprosy elimination target, leprosy hospitals were closed and leprosy clinics remained in 3 places. One of these is in our hospital as a specialized leprosy clinic and leprosy patients are followed here. The data were collected from the computer records and patient files. The study had a retrospective design and included patients who were hospitalized for a long time because of leprosy-related sequelae and social problems. Routine blood tests and examinations are performed on these patients at regular intervals, including thyroid tests and ultrasonography. In the present study, the computer records of the patients in the last 5 years were examined retrospectively, and the examinations were performed on those with thyroid-related data. All lepramatous leprosy patients whose thyroid function tests were checked were included in the study. The mean duration of the disease was found to be 45±14.1 years. A total of 21 patients were hospitalized during the study period. None of them were receiving drug therapy for leprosy. Patients has not receiving any medical treatment that would affect thyroid functions. The inclusion criteria for the study were the patient's being hospitalized in the leprosy clinic and thyroid data being available.

Age, gender, thyroid function test results, Thyroid-stimulating hormone (TSH), free triiodothyronine (FT3), free thyroxine (FT4) anti-thyroid peroxidase (Anti-Tpo), antithyroglobulin (Anti-Tg) and thyroglobulin (Tg) thyroid antibodies and thyroid ultrasonography results of the patients were analyzed. Thyroid diseases occur 5 times more frequently in women than in men in the general population. For this reason, the patients included in the study group were analyzed comparatively in terms of male and female gender.

Anti-TPO, anti-Tg, fT3, fT4 and thyroid-stimulating hormone (TSH) were studied with ECLIA (Electrochemiluminescence Immunoassay) (Roche Diagnostics GmbH, D-68298 Mannheim). TSH was measured with an analytical sensitivity of 0.005 µIU/mL. The Anti-Tg measurement range was considered as Anti-Tpo, Anti-Tg (normal, 0–115 IU/mL), Anti- Tpo (normal:0–34 IU/mL) and anti-TPO<35 ıu/mL negative. The thyroid of the patients was evaluated using a digital ultrasonography device (Philips EPIQ 7G) eL18-4 MHZ linear probe in the supine position with their heads in moderate hyperextension. The thyroid gland's size, parenchymal structure, presence of nodules, nodular characteristics, parenchyma and size (thickness x width x length x 0.52 = volume), echogenicity (hypoechoic, heterogeneous, isoechoic), vascularity (hypoechoic, mild, moderate and hypervascular), structure of the nodule and presence of lymphadenopathy were evaluated by a radiologist according to the Thyroid Imaging Reporting and Data Systems (TIRADS) Criteria. Thyroid nodules frequently occur in endemic areas due to insufficient iodine intake. Palpable nodules can be seen in 4 to 7 percent of the population, but nodules found incidentally by ultrasonographic examination can be found in 19 to 67 percent. The majority of thyroid nodules are asymptomatic.

Euthyroidism: It was defined as having TSH, FT3 and FT4 within the normal ranges and no symptoms in the subjects. TSH normal range is generally 0.27–4.2 uIU/mL U/mL. Subclinical hypothyroidism was defined as normal FT4 with a mildly elevated TSH. This biochemical finding may or may not be accompanied by mild symptoms of hypothyroidism. Subclinical hyperthyroidism is defined as a mildly suppressed TSH (generally still > 0.1 mIU/mL) in a patient without overt symptoms of hyperthyroidism. Overt hypothyroidism was defined as low FT4 (normal range 0.9–1.7 ng/dL) with elevated TSH, the sick euthyroid syndrome was defined as the presence of low FT3 (normal range 2–4.4 pg/mL) and FT4 along with normal TSH levels. Anti-TPO ABs is usually elevated in autoimmune thyroid disease, especially Hashimoto's Thyroiditis, postpartum thyroiditis, and Graves' Disease but anti-TPO ABs is also frequently found in euthyroid individuals.

Overt hyperthyroidism was defined as elevated FT4 with low TSH and subclinical hyperthyroidism as normal FT4 with suppressed TSH. Chronic thyroiditis is a condition characterized by slow-developing hypothyroidism, the growth of the thyroid gland depending on inflammatory cells or autoimmune atrophy with higher serum thyroid autoantibodies than normal and heterogeneous, hypoechogenic parenchymal pseudonodules on thyroid US [5].

## Statistical analysis

Statistical Package for Social Sciences for Windows (SPSS) 24.0 software was used for the analysis of the data obtained in the study. Frequency and percentage distribution analysis was used to determine the gender distribution of the patients who were included in the study. The mean and standard deviation values were analyzed to determine age, TSH, Free T3, T4 Anti-Tpo, Anti-TG and TG values of the participants. The independent Samples t-test was used to analyze the differences of TSH, T3, T4 etc. parameters of the patients in terms of gender and presence of nodules.

## Results

A total of 17 patients were included in the study. The ages of the patients ranged between 57 and 92, with a mean age of 75.12±9.89 years, and 10 patients (58.8%) were male. The mean TSH value of all patients was 1.78±1.23 µIU/mL and between 0.005 and 5.50. The mean FT3 (Free T3) value was 3.21±0.57 pg/ml, the mean FT4 (Free T4) was 1.19±0.23 ng/dl, the mean Anti-TPO was 12.27±10.63 IU/mL, the mean Anti-TG was 13.10±4.85 IU/mL, and the mean TG was 30.87±31.05 ng/ml

**Table 1. Shows the characteristics of the nodules according to gender.**

| Characteristics (number of patients with US, N = 14) | | |
|---|---|---|
| Nodule | N | % |
| Yes | 7 | 50 |
| No | 7 | 50 |
| Mean Nodule Size (cm) | 1.9±0.9 cm (0.7–3.4 cm) | |
| Mean Volume in Patients Without Nodule | Right Lobe | Left Lobe |
| | 2:41±1.8 cm$^3$ | 2.83±1.99 cm$^3$ |
| Mean Volume in Female Patients (cm$^3$) | Right Lobe | Left Lobe |
| | 4.68±1.58 (1.9–5.7) cm$^3$ | 4:44±1.99 (1.20–6.2) cm$^3$ |
| Mean Volume in Male Patients (cm$^3$) | 2.77±1.83(0.5–5.54) cm$^3$ | 3:16±1.77 (0.8–6.20) cm$^3$ |
| TIRADS Distribution of Nodules (N = 7) | TIRADS-2 | TIRADS-3 |
| | 4 Patients (57%) | 3 Patients (43%) |

(Independent Samples t-Test, *p<0.05; **p0.01)

When the patients were evaluated based on their thyroid functions, 14 (82.4%) had euthyroid, 1 (5.9%) had subclinical hypothyroidism, 1 female patient (5.9%) had hyperthyroidism and 1 (5.9%) had iodine deficiency. Although one patient had high TPO antibodies, the thyroid function tests were not clinically and ultrasonographically consistent with thyroiditis. Although the US of one male patient was consistent with chronic thyroiditis, thyroiditis findings were not detected in terms of clinical and laboratory results.

Thyroid US was performed on 14 of the patients, and the mean right lobe volume was found to be 2.41±1.8, and the left lobe volume was 2.83±1.99 in patients without nodules. In female patients, the mean right lobe volume was 4.68±1.58 (1.9–5.7) cm$^3$ and the mean left lobe volume was 4.44±1.99 (1.20–6.2) cm$^3$. In male patients, the mean right lobe volume was 2.77±1.83 (0.5–5.54) and the mean left lobe volume was 3.16±1.77 (0.8–6.20). Table 1 shows the characteristics of the nodules according to gender.

Nodules were noted in a total of 7 (50%) patients. Of these, 2 (29%) were male, and 5 (71%) were female. The nodules were multiple and bilateral in all patients. The mean nodule size was 1.9±0.9 cm (0.7–3.4 cm). When the nodules were evaluated according to TIRADS, they were found to be TIRADS-2 in 4 (57%) patients and TIRADS-3 (43%) in three patients. (Fig 1).

There were statistically significant differences between FT4, Anti-TPO and Anti-TG values in terms of gender. The FT4 (p:0.049<0.05), Anti-TPO (p:0.0450.05) and Anti-TG (p:0.0320.05) values of male patients were higher than those of female patients. Table 2 shows the comparison of TFT and thyroid antibodies according to gender including the sample size of male/female (M/F) participants.

Table 3 shows the comparison of TFT and USS findings of the thyroid nodules. There were statistically significant differences between TG values in terms of the presence of nodules. Accordingly, the TG (p:0.042<0.05) values of the patients with nodules were higher than those of the patients without nodules.

## Discussion

The structure and functions of the thyroid gland are affected in many diseases or may affect the course of the disease. There are few studies on the status or functions of the thyroid gland in leprosy patients, and the results differ. Our country is not endemic for leprosy [6]. Although we do not have any data at hand, the prevalence of leprosy is very low in our region and no new patients have been registered to the Leprosy Center in our hospital in the last 8 years.

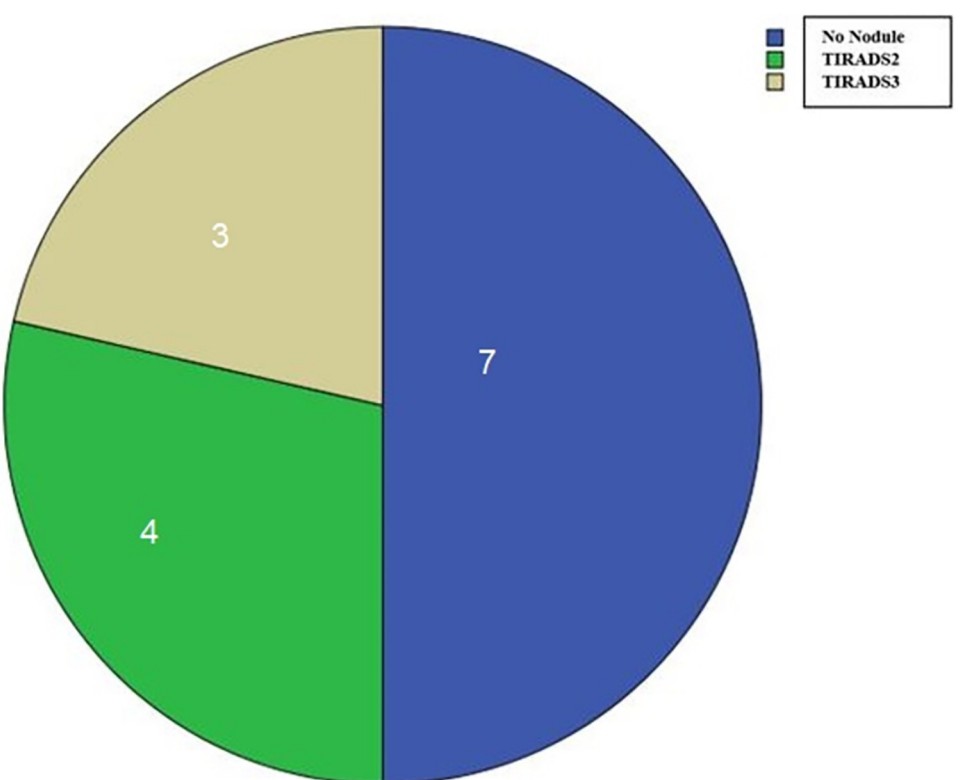

**Fig 1. The Pie chart shows that distribution of patient's thyroid ultrasonographic findings.**

Therefore, our sample size was limited to 17 patients. These patients consisted of advanced age patients due to the very low incidence of new leprosy.

The mean age of the patients was 75.12±9.89 years. In the normal population, structural and functional changes emerge in thyroid as age progresses. With increasing age, nodules develop in the thyroid gland, fibrosis and lymphocyte counts increase. The number and volume of follicles and the amount of colloid decrease. There are different study results regarding the total volume of the thyroid gland [7]. There are studies suggesting that the volume increases until the age of 40 in regions with iodine deficiency, and the volume decreases after the age of 40 in regions without iodine deficiency. The volume of a normal thyroid gland

**Table 2. Shows the comparison of TFT and thyroid antibodies according to gender, including the sample size of male/female participants.**

| Parameters | Gender | | | | Test Statistics | |
|---|---|---|---|---|---|---|
| | Female | | Male | | | |
| | $\bar{x}$ | s.d. | $\bar{x}$ | s.d. | t | p |
| TSH | 1.68 | 0.92 | 1.86 | 1.46 | -.290 | .776 |
| FT3 | 3.07 | 0.52 | 3.30 | 0.61 | -.797 | .438 |
| FT4 | 1.08 | 0.21 | 1.26 | 0.22 | -1.705 | **.049*** |
| Anti-TPO | 8.01 | 2.37 | 15.26 | 13.15 | -1.702 | **.045*** |
| Antithyroglobulin | 10.84 | 1.14 | 14.69 | 5.84 | -2.029 | **.032*** |
| Thyroglobulin | 29.00 | 19.12 | 32.18 | 38.25 | -.225 | .825 |

(Independent Samples t-Test, *p<0.05; **p0.01)

**Table 3. Shows the comparison of TFT and US findings of the thyroid nodules.**

| Parameters | Nodule | | | | Test Statistics | |
|---|---|---|---|---|---|---|
| | Yes | | No | | | |
| | $\bar{x}$ | s.d. | $\bar{x}$ | s.d. | t | p |
| TSH | 1.88 | 1.63 | 1.69 | 0.83 | .301 | .768 |
| FT3 | 3.37 | 0.72 | 3.06 | 0.38 | 1.091 | .292 |
| FT4 | 1.17 | 0.32 | 1.21 | 0.14 | -.353 | .729 |
| Anti-TPO | 12.02 | 7.45 | 12.50 | 13.31 | -.091 | .929 |
| Antithyroglobulin | 14.59 | 6.31 | 11.79 | 2.83 | 1.103 | .225 |
| Thyroglobulin | 40.07 | 41.02 | 22.69 | 17.26 | **1.665** | **.042*** |

(Independent Samples t-Test, *p<0.05; **p0.01)

without nodule varies in males and females. In publications, the thyroid volume in females has been reported as 18 mL and approximately 25 mL in males [8,9,10,11]. In leprosy patients without nodules, the mean right lobe volume was 2.41±1.8 cm$^3$, and the mean left lobe volume was 2.83±1.99 cm$^3$. When the patients with and without nodules were evaluated together, the mean right lobe volume was 4.68±1.58 (1.9–5.7) cm$^3$, and the mean left lobe volume was 4.44 ±1.99 (1.20–6.2) cm$^3$ in females. In males, the mean right lobe volume was 2.77±1.83 (0.5–5.54) cm$^3$ and the mean left lobe volume was 3.16±1.77 (0.8–6.20) cm$^3$.

The prevalence of thyroid nodules increases with age, and the rate of detection by ultrasonography can be 50% [7]. In our study, the mean size of the nodules was 19±09 (0.7–3.4) mm, and the nodule characteristics were benign among 14 patients with ultrasonographic examinations. No pathological examination was performed for nodules evaluated as TIRADS-2 and 3. The presence of benign nodules again suggests that leprosy and medical treatments for leprosy are not associated with malignancy in the thyroid. In the literature, the diagnosis of thyroid cancer in cases of thyroid nodules has been reported to be between 7% and 15% [12].

Researchers have conducted studies to examine thyroid functions in leprosy patients, and conflicting results have been found in previous studies regarding thyroid functions and diseases. These differences may be due to patient selection and measurement differences. Sehgal et al. analyzed thyroid functions with radioactive iodine in 17 patients with lepramatous leprosy. Except for only 2 patients, euthyroid was detected. They reported that thyroid functions were generally normal [13]. In their studies, Kheir et al. and Balybin found that serum T3 levels were high, which was not statistically significant, while Garg et al. found that the mean T3 levels were low. In patients with lepramatous leprosy, they found low mean serum T4 levels in correlation with the severity, duration of the disease and bacterial load. Balybin found no difference between the T4 levels of control and leprosy patients. Kheir and Balybin found that the mean serum TSH levels were statistically significantly higher compared to controls. Kheir hypothesized that leprosy patients might have subclinical hypothyroidism or it might be a variant of euthyroid sick syndrome [14,15,16].

Singh et al. found no thyroid dysfunction in leprosy patients and reported conflicting results with those of previous studies and suggested that it might be euthyroid sick syndrome [4]. Rolston et al. and Yumnam et al. reported normal serum levels of triiodothyronine (T3), thyroxine (T4) and thyroid-stimulating hormone (TSH) in leprosy [17,18,19].

In the present study, the mean levels of serum TSH, FT3, FT4, the most recent tests showing thyroid functions, were also within normal limits, and the incidence of hypothyroidism and hyperthyroidism was found to be similar to the population without leprosy. In addition, the Anti-Tpo, Anti-TG, and TG tests, which show autoimmune reaction in serum, were found to

be within normal limits in the patients. US, which reflects the structural characteristics of the thyroid tissue, was performed for the first time on leprosy patients, and the presence of nodules and suspicion of malignancy were determined at the same rate as the normal population. Since our region is endemic for thyroid diseasesthyroid sizes were found to be above normal. Since we did not perform histopathological examination, we do not know whether this size is related to the amyloid accumulation, which was observed in previous studies. The levels of FT3, FT4, TSH and anti-TG do not change with increasing age [7]. The mean FT3, FT4, TSH, Anti-Tpo, Anti-TG and TG values of our patients were found to be within normal limits. TG can be considered a marker of quantity, activity, and destruction of thyroid tissue and its concentration depends on the stimulation by thyrotropin (TSH) or TSH receptor autoantibodies. For this reason, TG levels increase in the bloodstream in hyperthyroidism, goiter, thyroid cancer derived from follicular epithelial cells, thyroid traumas, or destructive inflammation. It was found in the study that there was a statistically significant difference between the thyroglobulin levels of the patients with thyroid nodules and the thyroglobulin levels of the patients without nodules, which might be associated with these conditions.

Limitations: Our study has some limitations that the simple size was small because leprosy is not endemic in our country and a newly diagnosed cases are very few. In addition to there is no thyroid tests and thyroid US data of all patients with LL. The study was retrospective and the data of all patients different time period but all data include last 5 years.Another limitation is there is no comparison with the no LL patients as control group.

In conclusion, previous studies on this topic were generally conducted a few decades ago. The present study can give an opinion about thyroid functions and ultrasonographic characteristics in follow up of lepramatous lepra patients. Although leprosy is not a serious public healthcare concern in our region, there are regions in the world where leprosy still threatens public healthcare. For this reason, investigating whether there is a relationship between leprosy and thyroid disorders might contribute to the follow-up and treatment of leprosy patients. In the present study, no changes were detected in the thyroid functions and structures of the patients with Lepromatous Leprosy in a thyroid-endemic region. We think that prospective randomized studies that will include larger sample sizes are needed to determine whether there is a relationship between leprosy and thyroid disease. However, the study will shed light on the subject and further studies.

## Author Contributions

**Conceptualization:** Zeynep Ozkan, Ozkan Alatas.

**Data curation:** Zekiye Kanat, Ozkan Alatas, Zuhal Karaca Karagoz.

**Formal analysis:** Zeynep Ozkan.

**Investigation:** Zekiye Kanat.

**Methodology:** Zeynep Ozkan, Zuhal Karaca Karagoz.

**Project administration:** Zeynep Ozkan.

**Resources:** Ozkan Alatas.

**Supervision:** Zeynep Ozkan, Zuhal Karaca Karagoz.

**Validation:** Zuhal Karaca Karagoz.

**Visualization:** Ozkan Alatas.

**Writing – original draft:** Zeynep Ozkan.

**Writing – review & editing:** Zeynep Ozkan, Zuhal Karaca Karagoz.

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
