## [Decision Letter · Decision Letter 0]

31 Mar 2024

Dear Ozkan,

Thank you very much for submitting your manuscript "A RETROSPECTIVE ANALYSIS THYROID FUNCTION AND ULTRASONOGRAPHY IN A GROUP OF SUBJECTS WITH LEPROMATOUS LEPROSY IN TURKEY" for consideration at PLOS Neglected Tropical Diseases. As with all papers reviewed by the journal, your manuscript was reviewed by members of the editorial board and by several independent reviewers. In light of the reviews (below this email), we would like to invite the resubmission of a significantly-revised version that takes into account the reviewers' comments. 

We cannot make any decision about publication until we have seen the revised manuscript and your response to the reviewers' comments. Your revised manuscript is also likely to be sent to reviewers for further evaluation.

Sincerely,

Elsio A Wunder Jr, DVM, Ph.D.

Section Editor

Elsio Wunder Jr

Section Editor

Reviewer's Responses to Questions

**Key Review Criteria Required for Acceptance?**

**Methods**

-Are the objectives of the study clearly articulated with a clear testable hypothesis stated?

-Is the study design appropriate to address the stated objectives?

-Is the population clearly described and appropriate for the hypothesis being tested?

-Is the sample size sufficient to ensure adequate power to address the hypothesis being tested?

-Were correct statistical analysis used to support conclusions?

-Are there concerns about ethical or regulatory requirements being met?

Reviewer #1: -The objective of the study clearly articulated 

-the population didn't clearly described, how to confirm the study population were previously diagnosed with

lepramatous leprosy ( what type of diagnosed?)

- A sample size of 17 is relatively small. While it's possible to conduct analyses, including t-tests, with this sample size, the results may have limited generalizability to the population. Small sample sizes can lead to less precise estimates and may not adequately. With a small sample size, violations of these assumptions can have a larger impact on the validity of the results. With small sample sizes, there's an increased risk of both Type I (false positive) and Type II (false negative) errors. Researchers should be cautious when interpreting results and consider the potential for error due to the small sample size.

-Correct statistical analysis used to support conclusions but statistical analysis assumption is not explained well. 

-Ethical or regulatory requirements being met

Reviewer #2: In Materials and methods section it says "All patients had received any medication for leprosy and were follow up only." 

Does it mean that thyroid USG and blood tests were done before multi drug therapy (MDT) started for leprosy? Was MDT started after that? Or these patients were treated for leprosy before and testes were performed during follow ups? Please, clarify it.

**Results**

-Does the analysis presented match the analysis plan?

-Are the results clearly and completely presented?

-Are the figures (Tables, Images) of sufficient quality for clarity?

Reviewer #1: -The analysis presented match the analysis plan

- The results did not clearly and completely presented with the important variables ( not include all socio-demography , family history .... and others factors or variables 

- Include graphs in addition to tables

Reviewer #2: Yes, it is clear.

**Conclusions**

-Are the conclusions supported by the data presented?

-Are the limitations of analysis clearly described?

-Do the authors discuss how these data can be helpful to advance our understanding of the topic under study?

-Is public health relevance addressed?

Reviewer #1: - The conclusions supported by the data presented

- The limitations of analysis clearly not described all issues

-The authors discuss how the data can be helpful to advance our understanding of the topic under study 

- Public health relevance addressed properly

Reviewer #2: Yes, Conclusion is clear.

Limitation is described in discussion session. It would have been better to write the limitation in separate session.

**Editorial and Data Presentation Modifications?**

Reviewer #1: (No Response)

Reviewer #2: This can be accepted with minor revision as mentioned above.

**Summary and General Comments**

Reviewer #1: Requesting major revision by addressing increase sample size by using more than 5 years data if available, include important variables for description and analysis.

Reviewer #2: Well written paper. Can be accepted with minor revisions.

PLOS authors have the option to publish the peer review history of their article (what does this mean?). If published, this will include your full peer review and any attached files.

Reviewer #1: No

Reviewer #2: Yes: Indra Bahadur Napit
---

## [Decision Letter · Decision Letter 1]

8 Jul 2024

Dear Ozkan,

We are pleased to inform you that your manuscript 'A RETROSPECTIVE ANALYSIS THYROID FUNCTION AND ULTRASONOGRAPHY IN A GROUP OF SUBJECTS WITH LEPROMATOUS LEPROSY IN TURKEY' has been provisionally accepted for publication in PLOS Neglected Tropical Diseases.

Best regards,

Elsio A Wunder Jr, DVM, Ph.D.

Section Editor

Elsio Wunder Jr

Section Editor

Reviewer's Responses to Questions

**Key Review Criteria Required for Acceptance?**

**Methods**

-Are the objectives of the study clearly articulated with a clear testable hypothesis stated?

-Is the study design appropriate to address the stated objectives?

-Is the population clearly described and appropriate for the hypothesis being tested?

-Is the sample size sufficient to ensure adequate power to address the hypothesis being tested?

-Were correct statistical analysis used to support conclusions?

-Are there concerns about ethical or regulatory requirements being met?

Reviewer #2: (No Response)

**Results**

-Does the analysis presented match the analysis plan?

-Are the results clearly and completely presented?

-Are the figures (Tables, Images) of sufficient quality for clarity?

Reviewer #2: (No Response)

**Conclusions**

-Are the conclusions supported by the data presented?

-Are the limitations of analysis clearly described?

-Do the authors discuss how these data can be helpful to advance our understanding of the topic under study?

-Is public health relevance addressed?

Reviewer #2: (No Response)

**Editorial and Data Presentation Modifications?**

Reviewer #2: (No Response)

**Summary and General Comments**

Reviewer #2: No further comments.

PLOS authors have the option to publish the peer review history of their article (what does this mean?). If published, this will include your full peer review and any attached files.

Reviewer #2: No

---

## [Editor Report · Acceptance letter]

12 Jul 2024

Dear Ozkan,

We are delighted to inform you that your manuscript, "A RETROSPECTIVE ANALYSIS THYROID FUNCTION AND ULTRASONOGRAPHY IN A GROUP OF SUBJECTS WITH LEPROMATOUS LEPROSY IN TURKEY," has been formally accepted for publication in PLOS Neglected Tropical Diseases.

Best regards,

Shaden Kamhawi

co-Editor-in-Chief

Paul Brindley

co-Editor-in-Chief
